# The application of large language models in bariatric surgery: A scoping review

Ningjing Guo[1☉], Xuyan Li[1,2☉], Xiaoxue Li[1‡], Congmin Kang[1‡], Xiaoyan Gong[1‡], Xinyu Ji[1‡], Jie Zheng[iD][1*]

1 School of Nursing, Shanxi Medical University, Shanxi, China, 2 Heping Hospital Affiliated to Changzhi Medical College, Shanxi, China

☉ These authors contributed equally to this work and should be considered as co-first authors.
‡ These authors also contributed equally to this work.
* zhengjie@sxmu.edu.cn

## Abstract

### Background

Exploratory applications of large language models within the specialized field of metabolic and bariatric surgery have begun to emerge. Nevertheless, existing research remains fragmented, lacking comprehensive integration.

### Objective

To conduct a scoping review of studies on the application of large language models in the field of metabolic and bariatric surgery, aiming to provide a reference for clinical practice and future research.

### Methods

This scoping review adhered to the Joanna Briggs Institute methodological framework and followed the preferred reporting items for systematic reviews and meta-Analyses extension for scoping reviews (PRISMA-ScR) guidelines.PubMed, Web of Science, The Cochrane Library, Embase, CINAHL, CNKI, Wanfang, and VIP databases were searched for relevant studies, with the search timeframe from database inception to November 2025. The included literature was summarized and analyzed.

### Results

A total of 21 English-language studies were included. LLMs were primarily applied in scenarios such as patient education and information consultation, clinical decision support, and professional knowledge assessment. While LLMs performed well in information-provision tasks, they showed low consistency with expert opinions in complex clinical tasks such as individualized surgical recommendations.

**Data availability statement:** All relevant data are within the manuscript and its Supporting Information files.

**Funding:** The author(s) received no specific funding for this work.

**Competing interests:** The authors have declared that no competing interests exist.

Performance varied across different models, with GPT-4 generally demonstrating superior performance, and domain-specific models showing professional potential. Current research still faces challenges regarding information accuracy, readability, and clinical applicability.

## Conclusion

Large language models hold auxiliary potential in the field of metabolic and bariatric surgery, particularly for knowledge dissemination and patient education. However, their reliability in complex clinical decision-making remains limited. Future efforts should focus on conducting high-quality studies, advancing model specialization and standardized evaluation, and exploring safe and effective human-AI collaboration models.

## Introduction

The global prevalence of obesity continues to rise [1]. Obesity and its associated medical problems, such as type 2 diabetes, hypertension, and metabolic dysfunction-associated steatohepatitis (MASH), have emerged as major public health challenges that impact population health and increase socioeconomic and healthcare burdens [2,3]. Metabolic and bariatric surgery (MBS) has become an effective intervention for severe obesity, demonstrating not only significant weight loss but also improvement in metabolic parameters and a reduction in the incidence of obesity-related diseases. It is currently one of the most effective approaches for treating severe obesity and related metabolic disorders [4]. In recent years, MBS has been widely adopted in China and entered a phase of rapid development. According to statistics from the Chinese Obesity and Metabolic Surgery Database, the total annual number of MBS procedures in China increased to approximately 37,249 in 2025 [5] reflecting the growing clinical demand and vitality of the specialty.

However, alongside the sustained growth in surgical volume and the ongoing standardization of the specialty, metabolic and bariatric surgery continues to face a series of clinical challenges and practical pressures. For instance, in patient selection and surgical decision-making, choosing the most appropriate surgical technique remains a complex and contentious process [6]. Furthermore, due to the complexity of bariatric surgery and its long-term outcomes, patients require clear guidance and continuous support throughout the preoperative, intraoperative, and postoperative phases. This underscores the importance of effective communication, education, and accessible resources for improving patient empowerment and clinical outcomes [7]. These challenges constrain further enhancement in the precision of diagnosis and treatment, as well as the quality of long-term patient management, creating an urgent need for novel methods and tools to provide assistance.

In recent years, the rapid advancement of artificial intelligence (AI) technology, particularly represented by Large Language Models (LLMs), offers a new perspective and potential to address these challenges. In the medical field, LLMs have already

demonstrated significant potential in enhancing the quality of medical education, assisting clinical diagnosis, supporting decision-making, and promoting patient health management, among other areas [8]. Against this backdrop, exploratory applications of LLMs within the specialized field of metabolic and bariatric surgery have begun to emerge. Nevertheless, existing research remains fragmented, lacking comprehensive integration in terms of coverage across different application scenarios, systematic comparison of different model performances, and synthesis of common challenges and future directions faced by the technology. Therefore, this study aims to systematically review the relevant research on the application of LLMs in the field of metabolic and bariatric surgery through a scoping review methodology.

## Methods

### Type of review

This study was conducted according to the Joanna Briggs Institute methodology for scoping reviews [9]. Reporting adhered to the preferred reporting items for systematic reviews and meta-analyses extension for scoping reviews (PRISMA-ScR) (Fig 1) [10].

### Identifying the research question

The specific research questions that guided this review were as follows: (i) In which specific scenarios of bariatric surgery are LLMs applied, and how do they perform? (ii) What are the differences in performance among different LLM models when applied in bariatric surgery? (iii) What evidence do existing studies provide regarding the effectiveness of LLMs in bariatric surgery applications? (iv) What are the main challenges currently faced in these applications, and what are the future directions for development?

### Search strategy

A search was conducted in the electronic databases PubMed, Web of Science, The Cochrane Library, Embase, CINAHL, CNKI, Wan Fang, and VIP database, covering literature in both English and Chinese up to November 2025. Common search fields were used, employing a combination of subject headings and free-text keywords. References were also tracked throughout the review process. The full search strategy is provided in Table 1.

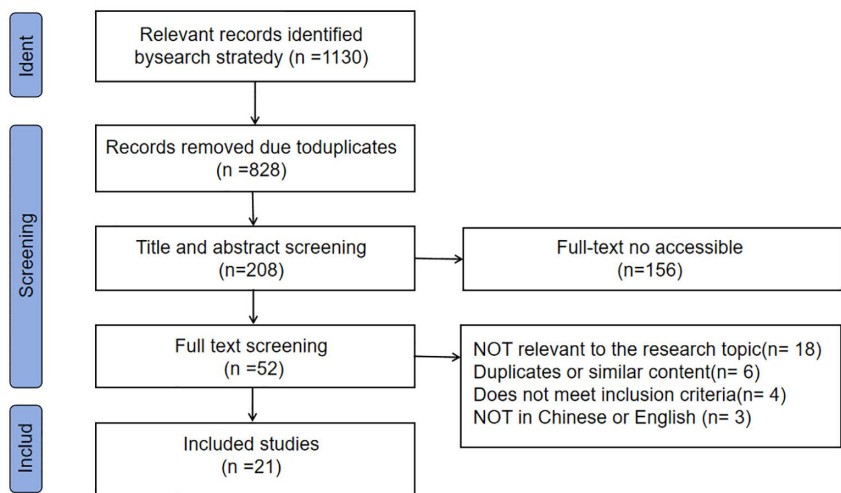

**Fig 1. PRISMA flow chart of the selection process.**

**Table 1. Search strategy used for each of the databases.**

| | |
|---|---|
| **PubMed** | |
| #1 | Large Language Models[Mesh] OR Artificial Intelligence[Mesh] |
| #2 | (Generative Pre-trained Transformer[tiab] OR ChatGPT[tiab] OR GPT[tiab] OR (Large[tiab] AND Language[tiab] AND Model[tiab]) OR LLM[tiab] OR LLMs[tiab] OR Gemini[tiab] OR DeepSeek[tiab] OR Copilot[tiab] OR LaMDA[tiab] OR LLaMA[tiab] OR BatGPT[tiab] OR Moss[tiab] OR Sora[tiab] OR Kimi[tiab] OR "Ernie Bot"[tiab] OR Qwen*[tiab] OR iFLYTEK Spark[tiab] OR ChatGLM*[tiab] OR Bard[tiab] OR Claude[tiab] OR Bing[tiab] OR Perplexity[tiab] OR mistral[tiab] OR Grok[tiab] OR PaLM[tiab] OR Chatsonic[tiab] OR Jasper[tiab] OR Generative Artificial Intelligence[tiab] OR Generative AI[tiab] OR Chatbot[tiab]) |
| #3 | #1 OR #2 |
| #4 | Bariatric Surgery[Mesh] OR Obesity, Morbid[Mesh] |
| #5 | (Bariatric[tiab] OR Metabolic Surgery[tiab] OR Weight Loss Surgery[tiab] OR Gastric Bypass[tiab] OR Sleeve Gastrectomy[tiab] OR Gastric Sleeve[tiab] OR Gastric Banding[tiab] OR Duodenal Switch[tiab]) |
| #6 | #4 OR #5 |
| #7 | #3 AND #6 |
| **Web of Science** | |
| #1 | TS=(Large Language Models OR Artificial Intelligence OR Generative Pre-trained Transformer OR ChatGPT OR GPT OR LLM OR LLMs OR Gemini OR DeepSeek OR Copilot OR LaMDA OR LLaMA OR BatGPT OR Moss OR Sora OR Kimi OR ErnieBot OR Qwen* OR iFLYTEK Spark OR ChatGLM* OR Bard OR Claude OR Bing OR Perplexity OR mistral OR Grok OR PaLM OR Chatsonic OR Jasper OR Generative Artificial Intelligence OR Generative AI OR Chatbot) |
| #2 | TS=(Bariatric Surgery OR Obesity, Morbid OR Bariatric OR Metabolic Surgery OR Weight Loss Surgery OR Gastric Bypass OR Sleeve Gastrectomy OR Gastric Sleeve OR Gastric Banding OR Duodenal Switch) |
| #3 | #1 AND #2 |
| **Cochrane** | |
| #1 | MeSH descriptor:[Large Language Models] explode all trees |
| #2 | MeSH descriptor: [Artificial Intelligence] explode all trees |
| #3 | #1 OR #2 |
| #4 | (Generative Pre-trained Transformer OR ChatGPT OR GPT OR LLM OR LLMs OR Gemini OR DeepSeek OR Copilot OR LaMDA OR LLaMA OR BatGPT OR Moss OR Sora OR Kimi OR ErnieBot OR Qwen* OR iFLYTEK Spark OR ChatGLM* OR Bard OR Claude OR Bing OR Perplexity OR mistral OR Grok OR PaLM OR Chatsonic OR Jasper OR Generative Artificial Intelligence OR Generative AI OR Chatbot):ti,ab,kw |
| #5 | #3 OR #4 |
| #6 | MeSH descriptor: [Bariatric Surgery] explode all trees |
| #7 | MeSH descriptor: [Obesity, Morbid] explode all trees |
| #8 | #6 OR#7 |
| #9 | (Bariatric OR Metabolic Surgery OR Weight Loss Surgery OR Gastric Bypass OR Sleeve Gastrectomy OR Gastric Sleeve OR Gastric Banding OR Duodenal Switch):ti,ab,kw |
| #10 | #8 OR#9 |
| #11 | #5 AND #10 |
| **Embase** | |
| #1 | 'large language model'/exp |
| #2 | 'artificial intelligence'/exp |
| #3 | #1 OR #2 |

*(Continued)*

**Table 1.** (Continued)

| PubMed | |
|---|---|
| #4 | generative pre-trained transformer:ti,ab,kw OR chatgpt:ti,ab,kw OR gpt:ti,ab,kw OR llm:ti,ab,kw OR llms:ti,ab,kw OR gemini:ti,ab,kw OR deepseek:ti,ab,kw OR copilot:ti,ab,kw OR lamda:ti,ab,kw OR llama:ti,ab,kw OR batgpt:ti,ab,kw OR moss:ti,ab,kw OR sora:ti,ab,kw OR kimi:ti,ab,kw OR 'ernie bot':ti,ab,kw OR qwen*:ti,ab,kw OR 'iflytek spark':ti,ab,kw OR chatglm*:ti,ab,kw OR bard:ti,ab,kw OR claude:ti,ab,kw OR bing:ti,ab,kw OR perplexity:ti,ab,kw OR mistral:ti,ab,kw OR grok:ti,ab,kw OR palm:ti,ab,kw OR chatsonic:ti,ab,kw OR jasper:ti,ab,kw OR generative artificial intelligence:ti,ab,kw OR generative ai:ti,ab,kw OR chatbot:ti,ab,kw |
| #5 | #3 OR #4 |
| #6 | 'bariatric surgery'/exp |
| #7 | 'morbid obesity'/exp |
| #8 | #6 OR #7 |
| #9 | bariatric:ti,ab,kw OR metabolic surgery:ti,ab,kw OR weight loss surgery:ti,ab,kw OR gastric bypass:ti,ab,kw OR sleeve gastrectomy:ti,ab,kw OR gastric sleeve:ti,ab,kw OR gastric banding:ti,ab,kw OR duodenal switch:ti,ab,kw |
| #10 | #8 OR #9 |
| #11 | #5 AND #10 |
| **CINAHL** | |
| S1 | MH Large Language Models OR MH Artificial Intelligence |
| S2 | TI ("Generative Pre-trained Transformer" OR "ChatGPT" OR "GPT" OR "LLM" OR "LLMs" OR "Gemini" OR "DeepSeek" OR "Copilot" OR "LaMDA" OR "LLaMA" OR "BatGPT" OR "Moss" OR "Sora" OR "Kimi" OR "ErnieBot" OR "Qwen*" OR "iFLYTEK Spark" OR "ChatGLM*" OR "Bard" OR "Claude" OR "Bing" OR "Perplexity" OR "mistral" OR "Grok" OR "PaLM" OR "Chatsonic" OR "Jasper" OR "Generative Artificial Intelligence" OR "Generative AI" OR "Chatbot") |
| S3 | S1 OR S2. |
| S4 | MH Bariatric Surgery OR MH Obesity, Morbid |
| S5 | TI ("Bariatric" OR "Metabolic Surgery" OR "Weight Loss Surgery" OR "Gastric Bypass" OR "Sleeve Gastrectomy" OR "Gastric Sleeve" OR "Gastric Banding" OR "Duodenal Switch") |
| S6 | S4 OR S5. |
| S7 | S3 AND S6. |

**China National Knowledge Infrastructure (CNKI) (Chinese)**

(Topic: "Large Language Model" OR "Artificial Intelligence" OR "Generative Pre-trained Transformer" OR "GPT" OR "ChatGPT" OR "DeepSeek" OR "Bard" OR "Bing" OR "Copilot" OR "LaMDA" OR "LLaMA" OR "BatGPT" OR "Moss" OR "Sora" OR "Kimi" OR "Generative Artificial Intelligence" OR "Chatbot") AND (Topic: "Bariatric Surgery" OR "Metabolic Surgery" OR "Obesity Surgery" OR "Gastric Bypass" OR "Sleeve Gastrectomy" OR "Gastric Banding" OR "Duodenal Switch")

**WANFANG DATA (Chinese)**

(Topic: "Large Language Model" OR "Artificial Intelligence" OR "Generative Pre-trained Transformer" OR "GPT" OR "ChatGPT" OR "DeepSeek" OR "Bard" OR "Bing" OR "Copilot" OR "LaMDA" OR "LLaMA" OR "BatGPT" OR "Moss" OR "Sora" OR "Kimi" OR "Generative Artificial Intelligence" OR "Chatbot") AND (Topic: "Bariatric Surgery" OR "Metabolic Surgery" OR "Obesity Surgery" OR "Gastric Bypass" OR "Sleeve Gastrectomy" OR "Gastric Banding" OR "Duodenal Switch")

**VIP database (Chinese)**

(Topic: "Large Language Model" OR "Artificial Intelligence" OR "Generative Pre-trained Transformer" OR "GPT" OR "ChatGPT" OR "DeepSeek" OR "Bard" OR "Bing" OR "Copilot" OR "LaMDA" OR "LLaMA" OR "BatGPT" OR "Moss" OR "Sora" OR "Kimi" OR "Generative Artificial Intelligence" OR "Chatbot") AND (Topic: "Bariatric Surgery" OR "Metabolic Surgery" OR "Obesity Surgery" OR "Gastric Bypass" OR "Sleeve Gastrectomy" OR "Gastric Banding" OR "Duodenal Switch")

## Literature inclusion and exclusion criteria

Inclusion criteria were determined according to the PCC (population, concept, context) principles [11] : (i) Participants (P): Involving clinical practice, patient management, or medical education in the field of metabolic and bariatric surgery;(ii) Concept (C): The core of the study involves the application of Large Language Models (LLMs), including but not limited to their development, deployment, evaluation, or comparison. Application forms include answering patient inquiries, generating educational materials, providing clinical decision support, etc.;(iii) Context (C): The application scenario is explicitly limited to metabolic and bariatric surgery. Study types are limited to original research such as quantitative studies, qualitative studies, and mixed-methods studies.Exclusion criteria: (i) Study content not directly related to bariatric surgery or the application of LLMs;(ii) Conference abstracts, commentaries, editorials, systematic reviews, literature reviews, case reports, study protocols, guidelines, or consensus statements;(iii) Literature for which the full text is unavailable, data cannot be extracted, or is non-peer-reviewed;(iv) Literature not published in Chinese or English.

## Study selection

After removing duplicates using EndNote X9 software, literature screening was performed by two researchers, strictly following the inclusion and exclusion criteria. The title and abstract were reviewed first, and the full text of studies potentially meeting the inclusion criteria was further examined. Any disagreements were discussed to reach an agreement, or a third party was consulted.

## Data extraction

The contents were extracted as follows: author, year, country, tool(s), application scenario category, specific application description, and main findings.

## Results

An initial search yielded 1,130 articles. After removing duplicates, screening titles and abstracts, and excluding articles without accessible full texts, 52 articles remained. Following full-text review, 21 English-language articles were ultimately included. The included studies were conducted in multiple countries, including the United States [12–20] (n = 9), China [21,22] (n = 2), Iran [23] (n = 1), Spain [24,25] (n = 2), Turkey [26,27] (n = 2), Canada [28,29] (n = 2), a multinational collaboration [30] (n = 1), Germany [31] (n = 1), and Brazil [32] (n = 1). The basic characteristics of the included studies are presented in Table 2.

## Application scenarios of LLMs in bariatric surgery

This review included a total of 21 studies. The application of Large Language Models in bariatric surgery primarily focused on the following scenarios: thirteen studies [12–16,19,21,22,26–28,31,32] concentrated on the patient education and information consultation scenario. This mainly involved providing patients with knowledge Q&A related to bariatric surgery, generating patient education materials, and optimizing text readability. This scenario represents the most widely explored area of LLM application, particularly demonstrating potential in improving the accessibility of disease-related information and patient engagement. Four studies [20,24,25,30] explored their application in clinical decision support, primarily focusing on recommendations for surgical techniques. However, consistency between LLMs and clinical guidelines or expert consensus in such tasks was low, with particularly limited performance in complex cases. Two studies [17,29] discussed the application of LLMs in bariatric surgery professional knowledge assessment and examinations, using them to simulate specialty board exam questions and evaluate the grasp of medical knowledge. Research indicated that LLMs performed well on standardized test questions but remained insufficient in clinical reasoning questions. One study [23] focused on medical image processing, attempting to use LLMs for bariatric surgery image recognition and generation. However, the

**Table 2. Basic characteristics of the literature for inclusion in this analysis (n=21).**

| Author | Year | Country | Tool(s) | Application scenario category | Specific application description | Main findings |
|---|---|---|---|---|---|---|
| Samaan et al. | 2023 | USA | ChatGPT-3.5 | Patient Education & Information | Model used to answer 151 common bariatric surgery questions to assess accuracy and reproducibility | ChatGPT-3.5 showed high accuracy (86.8% "completely accurate") in answering common questions and good reproducibility (90.7%). |
| Moazzam et al. | 2023 | USA | ChatGPT-3.5 | Patient Education & Information | Evaluated the quality of model responses to 30 bariatric surgery questions | Expert evaluation found ChatGPT-3.5 provided high-quality information; 50% supported integrating it into electronic health systems as a first-line response tool. |
| Aburumman et al. | 2024 | USA | ChatGPT-3.5 | Patient Education & Information | Compared model responses with hospital website answers to 8 ESG-related questions | Experts accurately identified AI-generated answers 54% of the time; overall, no significant differences were found between AI and website answers across assessed dimensions. |
| Srinivasan et al. | 2024 | USA | GPT-3.5, GPT-4, Bard | Patient Education & Information | Compared readability of bariatric surgery materials generated and simplified by three models | GPT-4 effectively simplified professional texts to a 6th-9th grade reading level with minimal loss of accuracy; Bard's simplification sometimes reduced answer completeness. |
| Samaan et al. | 2024 | USA | GPT-3.5, GPT-4 | Patient Education & Information | Compared accuracy and comprehensiveness of the two models in answering 151 bariatric surgery questions | GPT-4 matched GPT-3.5's accuracy while providing more comprehensive information in 36.4% of responses. |
| Sanders et al. | 2024 | USA | ChatGPT-4 | Knowledge Assessment & Exams | Tested model performance on metabolic and bariatric surgery simulated board examination questions | ChatGPT-4 achieved a 74.1% correct answer rate on specialty simulation questions, showing stable performance across different knowledge categories. |
| Ozmen et al. | 2025 | USA | Bariatric-SurgeryGPT (fine-tuned GPT-2) | Domain-Specific Model Development | Fine-tuned model with professional abstracts; evaluated its ability to generate specialty-specific content | The fine-tuned domain-specific model significantly outperformed the general GPT-2 model across all metrics for generating professional text, showing higher semantic relevance, terminology accuracy, and information completeness. |
| Annor et al. | 2025 | USA | Gemini, Copilot | Patient Education & Information | Evaluated the appropriateness and completeness of two models' responses to weight management guideline-related questions | Copilot provided more complete answers but exhibited guideline bias; Gemini was more cautious, sometimes refusing to answer or providing incomplete information. |
| Kahlon et al. | 2025 | USA | ChatGPT-4 | Clinical Decision Support | Compared model treatment recommendations with physician recommendations in five different bariatric surgery clinical scenarios | ChatGPT-4 showed good agreement with physician recommendations in simple clinical scenarios, but significant discrepancies were noted in complex cases. |
| Leng et al. | 2025 | China | ChatGPT-4 | Patient Education & Information | Designed 30 questions covering six domains of bariatric surgery; assessed the model's feasibility as a knowledge resource | ChatGPT-4 performed well in bariatric surgery Q&A, with 50% of responses receiving the highest score, but limitations included outdated criteria, lack of specificity, and suboptimal response structure. |
| Guo et al. | 2025 | China | ChatGPT-4, DeepSeek | Patient Education & Information | Compared the quality, reliability, and readability of patient education materials generated by two models | ChatGPT significantly outperformed DeepSeek in material quality and reliability; however, the readability of both models' outputs did not reach patient-friendly levels. |
| Mahjoubi et al. | 2025 | Iran | ChatGPT-4, DALL·E 3 | Medical Image Processing | Evaluated model accuracy in identifying surgical procedures and generating surgical illustrations | ChatGPT-4 and DALL·E 3 demonstrated low accuracy in identifying and generating bariatric surgery images, with issues such as missing anastomoses, incorrect anatomical depictions, and terminology errors in textual descriptions. |

*(Continued)*

**Table 2.** (Continued)

| Author | Year | Country | Tool(s) | Application scenario category | Specific application description | Main findings |
|---|---|---|---|---|---|---|
| **Lopez-Gonzalez et al.** | 2024 | Spain | ChatGPT-4 | Clinical Decision Support | Compared model-recommended surgical techniques with hospital algorithm decisions for 161 patients undergoing MBS | The match rate between ChatGPT-4 recommendations and the hospital algorithm decisions was only 34.16%, showing no significant correlation. |
| **Sanchez-Cordero et al.** | 2025 | Spain | ChatGPT | Clinical Decision Support | Evaluated consistency between model recommendations and clinical decisions for 283 primary bariatric surgery patients, pre- and post-exposure to scientific literature | Even after literature-based training, ChatGPT's recommendations showed low agreement with clinical decisions (25.8%). Its outputs aligned more closely with global surgical trends, while individual patient matching remained limited. |
| **Aksoy et al.** | 2025 | Turkey | ChatGPT-4.0, ChatGPT-Omni, Gemini | Patient Education & Information | Evaluated three models' performance on 720 One Anastomosis Gastric Bypass (OAGB) related clinical questions | ChatGPT-Omni demonstrated the best performance in answering OAGB-related clinical questions, with higher precision, recall, and F1-scores compared to the other models. |
| **Kumru Yildirim et al.** | 2025 | Turkey | ChatGPT-3.5 | Patient Education & Information | Evaluated consistency between model-provided postoperative nutrition advice for 27 questions and clinical guidelines | ChatGPT-3.5's postoperative nutrition advice was fully consistent with clinical guidelines in only 44.4% of cases; it showed weak personalization capability and made typical errors, such as misclassifying leafy greens as "high-protein food." |
| **Lee et al.** | 2024 | Canada | ChatGPT-4, Bing, Bard | Patient Education & Information | Evaluated appropriateness and readability of three models' responses to 36 clinical guideline-related questions | ChatGPT-4 had a significantly higher proportion of appropriate responses (85.7%) compared to Bard (74.3%) and Bing (25.7%), yet all remained in the "fairly difficult to read" range. ChatGPT-4 performed best in categories like surgical techniques and perioperative management. |
| **Lee et al.** | 2024 | Canada | ChatGPT-4, Bing, Bard | Knowledge Assessment & Exams | Tested the accuracy of three models on 200 multiple-choice questions in bariatric surgery | ChatGPT-4 achieved the highest overall accuracy (83.0%), followed by Bard (76.0%) and Bing (65.5%). ChatGPT-4 also had the highest correct rates in "Treatment & Procedures" (83.1%) and "Complications" (91.7%) categories, and left no questions unanswered. |
| **Jazi et al.** | 2023 | Multinational | ChatGPT-4 | Clinical Decision Support | Compared model surgical recommendations with multinational expert consensus in 10 clinical cases | Model recommendations aligned with majority expert opinion in only 30% of cases; in 40% of cases, the model gave inconsistent answers across two attempts (reproducibility 60%). |
| **Vedder et al.** | 2025 | Germany | GPT-4o | Patient Education & Information | Compared model vs. human expert answers to 200 real patient questions on accuracy, completeness, empathy, and patient satisfaction | Model response time (2.7 seconds) was significantly shorter than human experts (87.2 seconds), and answer length (607 characters) was significantly longer than human experts (262 characters). Patients rated model answers significantly higher for clarity, completeness, and empathy; 64.9% preferred model responses. |
| **Bigolin et al.** | 2025 | Brazil | ChatGPT-3.5, ChatGPT-4 | Patient Education & Information | Evaluated quality, empathy, safety, and clinical acceptability of two model versions answering 10 postoperative questions | Both versions scored >7 on average across all questions. ChatGPT-4 significantly outperformed v3.5 in "satisfaction" and "acceptability" (P < 0.05). Both versions demonstrated good empathetic capabilities. |

results showed low anatomical accuracy, indicating that they are not yet suitable for clinical or educational purposes. One study [18] explored the development of a domain-specific Large Language Model for bariatric surgery. The specialty model, built by fine-tuning a general-purpose model, outperformed the base model in professional text generation tasks, demonstrating the potential of vertical domain optimization.

## Performance differences among different LLMs

The performance of different LLMs in bariatric surgery tasks showed significant variation. Among the ChatGPT series models, GPT-4 demonstrated higher informational accuracy in most studies and outperformed its predecessor GPT-3.5 in providing detailed, contextualized responses [16]. However, this series of models commonly exhibited issues such as initially low readability [15]and knowledge update lag [21]. The performance of other general-purpose models like Gemini, Bard, and DeepSeek varied: Gemini demonstrated higher caution in some studies, sometimes refusing to answer sensitive medical questions directly, though this could also lead to insufficient completeness in its responses [19], DeepSeek slightly outperformed ChatGPT in text readability but lagged significantly behind in information quality and reliability [22], while Bard and Bing models showed unstable performance across multiple studies [28,29], particularly displaying low appropriateness in clinical advice generation tasks [28]. Furthermore, domain-fine-tuned models such as BariatricSurgery GPT excelled in the accuracy of professional terminology and semantic relevance [18], highlighting the potential of enhancing model specialization through vertical domain optimization. Overall, the differences in model performance depend not only on their underlying architecture and training data but are also closely related to the task type, prompt engineering, and evaluation criteria.

## Evaluation of LLM application effectiveness in bariatric surgery

In terms of accuracy, LLMs performed well in common knowledge Q&A tasks. For instance, the ChatGPT series achieved an accuracy rate of up to 86.8% on common bariatric surgery questions [12]. However, in clinical decision support tasks requiring clinical judgment, their agreement rates with expert opinions or real clinical decisions were typically below 40% [24,25,30], with particularly low matching rates for personalized surgical recommendations [20,24]. Readability remains a significant concern. Studies have pointed out that the average reading level of LLM-generated texts often exceeds the 6th to 8th-grade level recommended by the American Medical Association, mostly falling between the 9th-grade and college levels [15,22]. Although targeted prompts can simplify texts to some extent, bringing the readability level down to grades 6–9 [15], consistently meeting patient-friendly reading standards remains challenging, and some responses suffer from structural verbosity and excessive detail [21]. Regarding empathy and patient satisfaction, a few studies have conducted evaluations. One study found that GPT-4o received significantly higher patient ratings for answer clarity, completeness, and empathy, with 64.9% of patients expressing a preference for AI-generated responses [31]. Another study also noted that ChatGPT scored high on empathy, and patient satisfaction and acceptability of AI responses were good [32]. In terms of information quality, ChatGPT's response quality generally surpassed that of other models [13,22], but issues such as missing citations and lack of transparency regarding information sources remain prevalent. Domain-specific models demonstrated superior performance in the semantic relevance and professionalism of generated content [18]. In terms of efficiency, LLMs demonstrated a clear advantage, with an average response generation time of only a few seconds, significantly shorter than the time required for clinicians to draft similar content [31], suggesting their potential value in improving clinical workflow efficiency.

## Discussion

### Performance variation and capability boundaries across application scenarios

Current research indicates a clear divergence in the efficacy of LLMs across different application scenarios within bariatric surgery. In the realm of patient education and information consultation, LLMs, leveraging their robust natural language

   

generation capabilities, can provide patients with comprehensive and accurate knowledge-related responses about surgery. Multiple studies demonstrate that their response quality has reached a relatively high level, with some models achieving accuracy rates exceeding 85% in relevant evaluations [12]. This aligns with the findings of Goudrar et al.[33], suggesting that LLMs can serve as a beneficial supplementary resource for patients seeking information on bariatric surgery and possess the potential to become auxiliary clinical education tools.

However, in scenarios requiring clinical judgment, particularly for tasks involving personalized surgical plan recommendations and decision support, the performance of existing models is significantly constrained. Their alignment with clinical guidelines or expert consensus is generally low. Lopez-Gonzalez et al.[24] reported only 34.16% concordance between GPT-4 and hospital algorithm decisions. Several factors contributed: ChatGPT-4's knowledge is updated only until April 2023, it can only access open-access articles, and it explicitly states it is not designed for professional medical use. Sanchez-Cordero et al.[25] found that even after contextual training with 412 scientific articles, concordance improved only from 20.0% to 25.8%. Notably, ChatGPT tended to mirror the global "average" procedure distribution rather than tailoring recommendations to individual patients. The authors also noted that using a single center's practice as the gold standard may introduce bias, as surgical choices vary across centers. Kahlon et al.[20] observed only fair agreement between ChatGPT-4 and bariatric surgeons, with moderate inconsistency across two runs. Surgeons can integrate nuanced, patient-specific information that AI cannot fully weigh, highlighting a key limitation. Jazi et al.[30] reported that ChatGPT-4 matched expert consensus in only 30% of complex cases, and gave inconsistent answers in 40% of scenarios. The model failed to recognize critical patient-specific risk factors. This issue extends beyond bariatric surgery. Research by de Menezes Torres et al.[34] on the use of the large language model ChatGPT in oral and maxillofacial surgery highlights its limitations in handling complex clinical decisions and providing personalized recommendations for cases such as oral cancer and orthognathic surgery. In summary, current LLMs lack individualized, context-aware reasoning; exhibit output instability; have outdated knowledge bases and limited access to full evidence; and cannot handle multi-criteria trade-offs.

This underscores that while LLMs may be useful as educational tools, they are currently not reliable for autonomous surgical decision-making, especially in complex clinical contexts, highlighting the value of human expertise [26]. This performance variation clearly delineates the current capability boundaries of LLMs: they excel at processing and generating structured medical knowledge but currently lack a comprehensive and personalized perspective for tasks requiring the integration of multi-dimensional information, understanding complex clinical contexts, and performing dynamic reasoning, indicating fundamental limitations remain.

## Potential value and integration positioning in clinical practice

LLMs demonstrate multifaceted supportive potential in bariatric surgery. Firstly, they can significantly enhance clinical workflow efficiency, for instance, by automating the generation of patient education materials and rapidly answering common consultation questions, thereby freeing healthcare professionals from repetitive informational tasks. By improving answer efficiency under physician supervision while maintaining accuracy, LLMs can optimize doctors' time management and enhance patient satisfaction in bariatric care communication [31]. They can serve as supplementary resources for patient education, clinician assistance, and public health promotion [21]. Secondly, they facilitate the dissemination of health information. Particularly in contexts of unevenly distributed medical resources, they can provide patients with timely and reliable basic medical knowledge, helping to bridge service gaps caused by geographical or economic disparities. A study on LLM applications in ophthalmology showed that large language models like GPT-4 could assist ophthalmologists in distinguishing urgent from routine visits, improving remote ophthalmic triage in low-resource settings [35], thereby offering remote services to patients. More importantly, LLMs performed excellently on metabolic and bariatric surgery specialty certification simulation questions, achieving a correct answer rate of 74.1% with consistent performance across different knowledge categories [17], indicating AI's potential application value in specialty medical education and exam preparation. They can serve as effective adjuncts in medical education, aiding in the training of medical students and residents. It is

crucial to clarify that at this stage, LLMs should be positioned as "assistive tools" rather than "decision-making agents." Their core value lies in augmenting, not replacing, the professional judgment of clinical experts.

## Challenges and future directions

The foremost issues are insufficient information accuracy and timeliness. The trustworthiness and safety of their outputs remain inadequate, posing risks such as hallucination generation, outdated information, inconsistency with the latest guidelines, low readability, and potential biases. This aligns with the findings of Sanker et al.[36]. Secondly, LLMs lack a deep understanding of complex clinical contexts and struggle to handle patient cases involving individualized factors like multiple comorbidities or previous surgical history, limiting their utility in clinical decision support. Furthermore, model outputs often suffer from poor interpretability; the reasoning process behind their generated conclusions is opaque, making it difficult for clinicians to trace and verify the information basis.

These technical limitations are closely intertwined with broader ethical and safety considerations, which deserve detailed elaboration. First, data privacy and confidentiality are critical concerns, as noted in multiple studies; patient queries often contain sensitive health information, and inputting such data into public or unencrypted LLM interfaces risks unauthorized access or data breaches, especially because some models store conversations for retraining, and the lack of transparency regarding how platforms handle personal data further exacerbates these risks. Second, algorithmic bias may lead to unfair or inaccurate recommendations; LLMs trained on non-representative datasets may generate inappropriate recommendations, thereby exacerbating the risk of healthcare disparities. Third, liability and accountability remain unclear; the "black-box" nature of LLMs makes their reasoning process opaque, and when an LLM contributes to a clinical decision that results in an adverse event, responsibility is ambiguous—it is unclear whether the developer, the hospital, the supervising physician, or the user should be held accountable, as current legal frameworks lack clear guidance for LLM-assisted surgical decisions. Fourth, over-reliance on LLMs may negatively affect the patient-clinician relationship, as patients might reduce trust in human providers or delay necessary consultations. Finally, the most frequently cited issue is the "hallucination" phenomenon—LLMs tend to generate responses with high confidence and coherent structure even when the content is factually incorrect, which may mislead patients and lead uninformed users to adopt erroneous and potentially dangerous advice. Taken together, these risks emphasize that LLMs are not yet mature and must currently be positioned as supportive tools under human supervision, not autonomous advisors.

Future development directions should include: integrating AI with human healthcare professionals as a key pathway to optimizing patient care [37], establishing evaluation standards oriented towards real-world clinical scenarios, exploring safe and effective human-AI collaborative workflows, and constructing comprehensive ethical governance frameworks. Specifically, future research should focus on three interconnected priorities. First, performance validation and multidimensional evaluation. Given the low consistency of LLMs in complex clinical decisions and their tendency to generate hallucinations or outdated information, rigorous and continuous evaluation and improvement are needed, including testing different models and prompt strategies. Special attention should be paid to reducing hallucinations, improving reproducibility, and enabling individualized recommendations. Assessment frameworks must systematically incorporate core metrics such as accuracy, reproducibility, readability, bias detection, and clinical utility, with standardized evaluation protocols. Second, development of human-AI collaboration models. Research should use various data sources to explore optimal integration patterns that balance efficiency and safety. Different collaboration models should be tested continuously across diverse simulated clinical scenarios, evaluating their impact on decision quality, workflow efficiency, clinician acceptance, and patient outcomes. Third, establishment of ethical and legal governance frameworks. Clear operational guidelines are urgently needed: for data privacy, secure handling of patient queries and preventing unauthorized use of data for other purposes; for liability allocation, defining responsible parties when LLM suggestions contribute to adverse events and establishing a clear accountability framework; and for informed consent, disclosing AI use to patients. Only through the

synergistic advancement of technological innovation, clinical validation, and regulatory management can the responsible application and sustainable development of LLMs in this field be achieved.

## Conclusions

Large Language Models have demonstrated potential as auxiliary informational tools and communication mediators in bariatric surgery, showing particular value in enhancing information accessibility, supporting doctor-patient communication, and medical education. However, their current capabilities remain confined to structured knowledge transmission and are not yet reliable for clinical tasks requiring professional judgment, personalized decision-making, and comprehension of complex contexts. Future development should focus on the specialized optimization of models, standardization of evaluation systems, and exploration of clinical integration pathways. This will facilitate the deep integration of AI with bariatric surgery under the premise of ensuring safety, reliability, and equity.

## Supporting information

**S1 Checklist. Preferred reporting items for systematic reviews and meta-analyses extension for scoping reviews (PRISMA-ScR) checklist.**
(DOCX)

## Author contributions

**Data curation:** Ningjing Guo, Xuyan Li, Xiaoxue Li, Congmin Kang, Xiaoyan Gong, Xinyu Ji, Jie Zheng.

**Writing – original draft:** Ningjing Guo, Xuyan Li.

**Writing – review & editing:** Ningjing Guo, Xuyan Li, Jie Zheng.

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
