## [Decision Letter · Decision Letter 0]

27 Apr 2026

PONE-D-26-11116The Application of Large Language Models in Bariatric Surgery: A Scoping ReviewPLOS One

Dear Dr. Zheng,

Thank you for submitting your manuscript to PLOS ONE. After careful consideration, we feel that it has merit but does not fully meet PLOS ONE’s publication criteria as it currently stands. Therefore, we invite you to submit a revised version of the manuscript that addresses the points raised during the review process.

We look forward to receiving your revised manuscript.

Kind regards,

Hongyang Ma, Ph.D., D.D.S

Guest Editor

PLOS One

Journal Requirements:

Reviewers' comments:

Reviewer's Responses to Questions

**Comments to the Author**

1. Is the manuscript technically sound, and do the data support the conclusions?

Reviewer #1: Yes

2. Has the statistical analysis been performed appropriately and rigorously? 

Reviewer #1: Yes

3. Have the authors made all data underlying the findings in their manuscript fully available?

Reviewer #1: Yes

4. Is the manuscript presented in an intelligible fashion and written in standard English?

Reviewer #1: Yes

5. Review Comments to the Author

Reviewer #1: This article is a comprehensive scoping review supported by an extensive literature review. Analyzing 21 English-language studies, it reveals the applications of LLMs in areas such as patient education, information counseling, clinical decision support systems, and professional knowledge assessment.

The study fills a significant gap in the field by exploring the potential use of LLMs in surgical applications. The scoping review methodology ensures a comprehensive approach to the subject, systematically presenting key findings and applications from the existing literature. However, the following points require improvement:

1. The consistency of LLMs in clinical decision-making processes was found to be low. This issue needs to be discussed in more detail.

2. The section on recommendations for future research could be more comprehensive. Specifically, areas requiring further study should be identified.

3. Providing more detail on the ethical and safety risks associated with the use of LLMs would be beneficial in addressing concerns in this area.

The article presents important findings and contributes to the field. Correcting the weaknesses mentioned above will improve the publishability of the article. It is important for the authors to review the work taking the suggested corrections into consideration. The article is considered suitable for publication after the necessary corrections have been made.

6. PLOS authors have the option to publish the peer review history of their article (what does this mean?). If published, this will include your full peer review and any attached files.

Reviewer #1: **Yes:** Mustafa Cem Algin

---

## [Author Response · Author response to Decision Letter 1]

8 May 2026

Dear Editor and Reviewers,

We sincerely appreciate the time and expertise invested in reviewing our manuscript 【PONE-D-26-11116. Thank you for your constructive feedback, which has significantly strengthened the quality of this work. We have carefully addressed all comments and revised the manuscript accordingly. Below, we provide a point-by-point response to each suggestion, with corresponding revisions highlighted in the tracked-changes version of the manuscript.

Journal Requirements:

1.“Please ensure that your manuscript meets PLOS ONE's style requirements, including those for file naming. ”

Response: We have revised the manuscript to fully comply with PLOS ONE’s formatting guidelines. Should any formatting inconsistencies persist despite our rigorous checks using the PLOS ONE templates, we remain fully open to implementing additional corrections as directed by the editorial office. A 24-hour revision turnaround is guaranteed upon request.

2.“Please include captions for your Supporting Information files at the end of your manuscript, and update any in-text citations to match accordingly.”

Response: 【line 523】

①　A dedicated paragraph titled "Supporting Information " has been inserted after the References section.

②　Captions for all Supporting Information files (e.g., S1 Table) have been added under the "Supporting Information" section at the end of the manuscript.

③　In-text citations to Supporting Information have been standardized

3.If the reviewer comments include a recommendation to cite specific previouslypublished works, please review and evaluate these publications to determine whetherthey are relevant and should be cited. There is no requirement to cite these works unless the editor has indicated otherwise.

We thank the editor for this reminder. We have carefully reviewed all comments from the reviewer and confirm that no specific recommendation was made to cite any particular previously published work. Therefore, no additional citations have been added in response to such a request. Nevertheless, we have thoroughly examined our reference list to ensure its completeness, correctness, and relevance, and we have confirmed that no retracted articles are cited. If the editor or any reviewer recommends specific references for inclusion, we will gladly evaluate and incorporate them as appropriate.

4.“Please review your reference list to ensure that it is complete and correct. If you have cited papers that have been retracted, please include the rationale for doing so in the manuscript text, or remove these references and replace them with relevant current references. Any changes to the reference list should be mentioned in the rebuttal letter that accompanies your revised manuscript. If you need to cite a retracted article, indicate the article’s retracted status in the References list and also include a citation and full reference for the retraction notice.”

We sincerely appreciate the reviewer's meticulous feedback regarding the integrity of our reference list. In accordance with the comment, we have thoroughly re-examined all cited references and confirm that no retracted articles were included in the manuscript.

Additional Editor Comments :

1. The consistency of LLMs in clinical decision-making processes was found to be low. This issue needs to be discussed in more detail.

Response: We thank the reviewer for raising this critical issue. We have substantially expanded the discussion on the low consistency of large language models (LLMs) in clinical decision making, as requested. Specifically, we have added a detailed analysis in the Discussion section (please see lines 238–259 in the revised manuscript, with tracked changes).

Lopez-Gonzalez et al.[24] reported only 34.16% concordance between GPT-4 and hospital algorithm decisions. Several factors contributed: ChatGPT-4’s knowledge is updated only until April 2023, it can only access open-access articles, and it explicitly states it is not designed for professional medical use. Sanchez-Cordero et al.[25] found that even after contextual training with 412 scientific articles, concordance improved only from 20.0% to 25.8%. Notably, ChatGPT tended to mirror the global “average” procedure distribution rather than tailoring recommendations to individual patients. The authors also noted that using a single center’s practice as the gold standard may introduce bias, as surgical choices vary across centers. Kahlon et al.[20] observed only fair agreement between ChatGPT-4 and bariatric surgeons, with moderate inconsistency across two runs. Surgeons can integrate nuanced, patient-specific information that AI cannot fully weigh, highlighting a key limitation. Jazi et al.[30] reported that ChatGPT-4 matched expert consensus in only 30% of complex cases, and gave inconsistent answers in 40% of scenarios. The model failed to recognize critical patient-specific risk factors. This issue extends beyond bariatric surgery.

In summary, current LLMs lack individualized, context-aware reasoning; exhibit output instability; have outdated knowledge bases and limited access to full evidence; and cannot handle multi-criteria trade-offs.

2. The section on recommendations for future research could be more comprehensive Specifically, areas requiring further study should be identified.

Response: We thank the reviewer for this valuable comment. We have revised and expanded the future research recommendations in the revised manuscript. In this expanded section, we clearly identify three interconnected priorities for future research .(please see lines 332–351 in the revised manuscript, with tracked changes).

Specifically, future research should focus on three interconnected priorities. First, performance validation and multidimensional evaluation. Given the low consistency of LLMs in complex clinical decisions and their tendency to generate hallucinations or outdated information, rigorous and continuous evaluation and improvement are needed, including testing different models and prompt strategies. Special attention should be paid to reducing hallucinations, improving reproducibility, and enabling individualized recommendations. Assessment frameworks must systematically incorporate core metrics such as accuracy, reproducibility, readability, bias detection, and clinical utility, with standardized evaluation protocols. Second, development of human AI collaboration models. Research should use various data sources to explore optimal integration patterns that balance efficiency and safety. Different collaboration models should be tested continuously across diverse simulated clinical scenarios, evaluating their impact on decision quality, workflow efficiency, clinician acceptance, and patient outcomes. Third, establishment of ethical and legal governance frameworks. Clear operational guidelines are urgently needed: for data privacy, secure handling of patient queries and preventing unauthorized use of data for other purposes; for liability allocation, defining responsible parties when LLM suggestions contribute to adverse events and establishing a clear accountability framework; and for informed consent, disclosing AI use to patients.

3. Providing more detail on the ethical and safety risks associated with the use of LLMs would be beneficial in addressing concerns in this area.

Response: We appreciate this valuable suggestion. We have added a dedicated, detailed paragraph on ethical and safety risks in the Discussion section. (please see lines 305–327 in the revised manuscript, with tracked changes).

These technical limitations are closely intertwined with broader ethical and safety considerations, which deserve detailed elaboration. First, data privacy and confidentiality are critical concerns, as noted in multiple studies; patient queries often contain sensitive health information, and inputting such data into public or unencrypted LLM interfaces risks unauthorized access or data breaches, especially because some models store conversations for retraining, and the lack of transparency regarding how platforms handle personal data further exacerbates these risks. Second, algorithmic bias may lead to unfair or inaccurate recommendations; LLMs trained on non representative datasets may generate inappropriate recommendations, thereby exacerbating the risk of healthcare disparities. Third, liability and accountability remain unclear; the “black box” nature of LLMs makes their reasoning process opaque, and when an LLM contributes to a clinical decision that results in an adverse event, responsibility is ambiguous—it is unclear whether the developer, the hospital, the supervising physician, or the user should be held accountable, as current legal frameworks lack clear guidance for LLM assisted surgical decisions. Fourth, over reliance on LLMs may negatively affect the patient clinician relationship, as patients might reduce trust in human providers or delay necessary consultations. Finally, the most frequently cited issue is the “hallucination” phenomenon—LLMs tend to generate responses with high confidence and coherent structure even when the content is factually incorrect, which may mislead patients and lead uninformed users to adopt erroneous and potentially dangerous advice. Taken together, these risks emphasize that LLMs are not yet mature and must currently be positioned as supportive tools under human supervision, not autonomous advisors.

Supplementary explanation:

We thank the editor for the clarification regarding figure technical validation. We confirm full compliance with the figure formatting requirements through the following actions:

① Fig 1 (Fig1.tif) has been successfully validated via the NAAS pre flight process.

② The NAAS validated Fig1.tif will replace the original figure file in the resubmitted manuscript.

2. Data Availability

Our study is a scoping review, not an original study generating new primary data. The minimal dataset required to replicate our findings—specifically, the key characteristics and results of the 21 included studies—is fully presented in Table 2 and the main text of the manuscript. No additional raw data were generated or analyzed beyond what is already reported.

Accordingly, we have updated the Data Availability Statement in the manuscript to read: " All relevant data are within the paper and its Supporting Information files. As this is a review article, no new primary data were generated."

We are grateful for the opportunity to improve our study through this revision. All changes have been thoroughly cross-checked for compliance with PLOS ONE’s guidelines, including figure formatting (validated via NASS), and reference integrity. Should additional clarifications or refinements be required, we remain fully committed to addressing them promptly.

Thank you once again for your invaluable input.

Sincerely,

Jie Zheng

Nursing College, Shanxi Medical University

zhengjie@sxmu.edu.cn

Tel: +86 18247758920

---

## [Decision Letter · Decision Letter 1]

19 May 2026

The Application of Large Language Models in Bariatric Surgery: A Scoping Review

PONE-D-26-11116R1

Dear Dr. Jie Zheng,

We’re pleased to inform you that your manuscript has been judged scientifically suitable for publication and will be formally accepted for publication once it meets all outstanding technical requirements.

Kind regards,

Hongyang Ma, Ph.D., D.D.S

Guest Editor

PLOS One

Additional Editor Comments (optional):

Reviewers' comments:

Reviewer's Responses to Questions

**Comments to the Author**

1. If the authors have adequately addressed your comments raised in a previous round of review and you feel that this manuscript is now acceptable for publication, you may indicate that here to bypass the “Comments to the Author” section, enter your conflict of interest statement in the “Confidential to Editor” section, and submit your "Accept" recommendation.

Reviewer #1: All comments have been addressed

2. Is the manuscript technically sound, and do the data support the conclusions?

Reviewer #1: Yes

3. Has the statistical analysis been performed appropriately and rigorously? 

Reviewer #1: No

4. Have the authors made all data underlying the findings in their manuscript fully available?

Reviewer #1: Yes

5. Is the manuscript presented in an intelligible fashion and written in standard English?

Reviewer #1: Yes

6. Review Comments to the Author

Reviewer #1: (No Response)

7. PLOS authors have the option to publish the peer review history of their article (what does this mean?). If published, this will include your full peer review and any attached files.

Reviewer #1: **Yes:** Mustafa Cem Algin

---

## [Editor Report · Acceptance letter]

PONE-D-26-11116R1

PLOS One

Dear Dr. Zheng,

I'm pleased to inform you that your manuscript has been deemed suitable for publication in PLOS One. Congratulations! Your manuscript is now being handed over to our production team.

Kind regards,

on behalf of

Dr. Hongyang Ma

Guest Editor

PLOS One